# Wear and Subsurface Stress Evolution in a Half-Space under Cyclic Flat-Punch Indentation

Javier M. Juliá 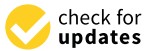 and Luis Rodríguez-Tembleque * 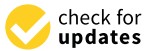

Department of Continuum Mechanics and Structures, Escuela Técnica Superior de Ingeniería,
Universidad de Sevilla, Camino de los Descubrimientos s/n, 41092 Sevilla, Spain
* Correspondence: luisroteso@us.es

**Abstract:** Wear is a tremendously important phenomenon, which takes place on the surfaces of two solids in contact under cyclic loads and constitutes one of the most-significant ways of failure for mechanical elements. However, it is not the only source of failure in contacting solids. The subsurface stresses should also be considered, due to the fatigue and crack initiation problems. Nevertheless, these stresses (i.e., their maximum values and distributions) evolve with the solids' surface wear (i.e., with the load cycles) and also depend on the friction intensity. Therefore, their evolution should be properly computed to predict failures in mechanical elements under wear conditions. This work focused on the study of the evolution of the surface wear and the subsurface stress distributions generated—in an elastic half-space—by a cylindrical flat-ended punch, under cyclic indentation loading (i.e., radial fretting wear conditions). Based on a numerical scheme recently presented by the authors, this is the first time that, for this contact problem, the surface wear and subsurface stress distribution (i.e., maximum value and its location)—and its evolution—were simultaneously analyzed when orthotropic friction and fretting wear conditions were considered. The studies presented in this work were developed for purely elastic contact assumptions.

**Keywords:** subsurface stress; wear; orthotropic friction; fretting wear; contact mechanics; cylindrical flat punch; flat-ended contact; cyclic indentation



## 1. Introduction

Friction and wear are inherent phenomena in the mechanical contact between two elastic bodies. Mechanical contact usually takes place in regions where forces are transmitted between two machine—or structural—components, i.e., in regions of joints, in bearings, in the contact region of a railway track and wheel, etc. Therefore, very high stresses appear at the surface—and subsurface—contact regions of these mechanical components. It should be noted that failure by induced subsurface stresses occurs when these stresses exceed the elastic limits according to a given/pertinent failure criterion. The failure is usually due to fatigue spalling/pitting [1], and that failure occurs when such limiting stresses coincide with subsurface flaws created during manufacturing such as pores, inclusions, cracks, etc. According to [2,3], the behavior of the subsurface stresses allows us to predict cracks' nucleation and expansion through the solid. However, that stress depends on the friction intensity and, under dynamic or cyclic loading, evolves the surface wear (i.e., with the load cycles).

The surface wear problem has been studied for more than seventy years (i.e., since the pioneering works of Holm [4] and Archard [5]). Since then, many theoretical and computing formulations have been proposed to predict surface wear under several contact conditions. The theoretical works of Galin et al. [6,7], Kovalenko et al. [8,9], Kragelsky [10], and Komogortsev [11] led to the fundamental works of Hills et al. [12,13] and Goryacheva et al. [14] in fretting wear problems. Olofsson et al. [15], Enblom and Berg [16], Telliskivi [17], and Hegadekatte et al. [18] proposed different solutions for rolling contact

problems (e.g., for pin-on-disc and twin-disc systems) under sliding wear conditions. More recently, Argatov et al. [19–22] and Di Puccio and Mattei [23,24] presented additional analytical solutions for sliding wear contact problems and Popov [25] and Cubillas et al. [26] for fretting wear.

Numerical solutions to compute surface wear have also been developed for the last thirty years. The pioneering works of Johansson [27] and Strömberg et al. [28–30] made it possible to solve contact problems under fretting wear or sliding wear conditions. These formulations—based on the finite element method (FEM)—were extended to thermoelastic contact problems in [31,32], to more realistic fretting wear problems in [33–36], and to study sliding wear problems in [37]. Recent works in the finite element context have focused on non-matching mesh schemes (see [38,39]). Numerical schemes based on the boundary element method (BEM)—or on the influence coefficient method (ICM)—were also presented to compute surface wear in contacting elements under: sliding wear [40–43], fretting wear [44–48], or rolling contact [49,50] conditions. Moreover, the atomistic simulations developed in the works of Aghababaei et al. [51,52] revealed how important the numerical simulations and modeling are becoming to explore wear processes.

Regarding the evaluation of the subsurface stresses, several works have been carried out since the pioneering works [53–56] or the monograph [57] were presented. Some of these recent studies were focused on the elastic line contact problem under dry contact conditions [58] for layered solids [59,60] or for an inhomogeneous elastic medium [61]. Subsurface stresses under lubricated elastic line contact conditions were analyzed in [62–64]. For point contact conditions, a comprehensive analysis of the subsurface stresses caused by a Hertzian ellipsoidal pressure distribution was provided by Sackfield and Hills in [65,66] and, more recently, by Greenwood [67]. The analysis of the subsurface stresses caused by any arbitrary pressure distribution was presented by Johns-Rahnejat and Gohar in [68]. The interest in the subsurface stress distributions caused by contact and how they can be affected by friction intensity [2,69,70] or wear [71] has drawn the attention of some researchers in recent years.

In this context, this work studied the evolution of the surface wear and the subsurface stress distributions generated—in an elastic half-space—by a vertically loaded cylindrical flat-ended punch under cyclic normal indentation loading (i.e., conformal contact and radial fretting wear conditions). This problem has been studied for many years, since it is widely present in many mechanical components. The stress distribution caused by a cylindrical flat-ended punch was tackled by Sneddon et al. [72,73] and later by Barquins and Maugis [74], who observed a stress singularity at the edge of the contact region. Later on, these solutions were collected in the fundamental books of Fischer-Cripps [75] and Popov et al. [76], together with the contact pressure distribution. However, to the best of the authors' knowledge, the surface wear and the subsurface stress distribution (i.e., maximum value and its location)—and their evolution—have not been simultaneously computed and analyzed for this problem. Therefore, this work analyzes—for the first time—the simultaneous evolution of the surface wear and the subsurface stress distribution caused by the indentation of a flat-ended cylindrical punch over an elastic half-space under orthotropic friction and radial fretting wear conditions. Moreover, this is the first time that the BEM was applied to analyze this indentation problem under these tribological contact conditions. For this purpose, the computing framework presented in [71] was extended to analyze the indentation of a flat-ended cylindrical punch over an elastic half-space. That computational scheme was based on the BEM (or ICM) [77–79] to obtain the influence coefficients of the contacting solids. The frictional contact problem was solved using an augmented Lagrangian formulation (see [80–82]). Then, the subsurface stresses in the contacting solids were computed according to Liu and Wang [83]. This scheme allowed us to study several numerical examples, where the influence of—orthotropic—wear and friction conditions were analyzed. All these numerical studies were developed under purely elastic contact assumptions.

Finally, the structure of this manuscript is organized as follows. After the Introduction, Section 2 presents the kinematic equations for the frictional contact problem including wear. Section 3 focuses on the BEM approximation, which makes it possible to obtain the relations between the surface displacements and the contact tractions. Section 4 defines the contact and wear—orthotropic—laws, and Section 5 presents the equations to compute the subsurface stress distributions. The numerical solution is briefly presented in Section 6. Then, Section 7 presents the analyses, and finally, the concluding remarks are compiled in Section 8.

## 2. Frictional Contact Kinematics Including Wear

The kinematic description of two elastic bodies ($\Omega^\alpha$, $\alpha = A, B$) contacting in common area $\Gamma_c$ needs to define the position $\mathbf{x}$ of each solid particle of these two bodies relative to a coordinate system $Oxyz$. In that system, $z$ points are normally oriented and $\{x, y\}$ are tangentially directed (see Figure 1). Then, the normal gap between these two solids' surfaces can be defined as

$$g_n(\mathbf{x}, \tau) = g_g - g_{n,o}(\tau) + \omega(\mathbf{x}, \tau) + u_n(\mathbf{x}, \tau), \tag{1}$$

where $g_g$ is the geometric gap, $g_{n,o}(\tau)$ is the rigid body approach at a certain pseudo-time instant ($\tau$) and $u_n$ is the relative surface normal displacement: $u_n(\mathbf{x}, \tau) = u_z^{(A)}(\mathbf{x}, \tau) - u_z^{(B)}(\mathbf{x}, \tau)$.

The tangential slip displacements are defined as

$$\mathbf{g}_t(\mathbf{x}, \tau) = \mathbf{g}_{t,o}(\tau) + \mathbf{u}_t(\mathbf{x}, \tau), \tag{2}$$

where $\mathbf{g}_{t,o}(\tau)$ is the rigid body tangential slip displacements and $\mathbf{u}_t(\mathbf{x}, \tau) = \mathbf{u}_t^{(A)}(\mathbf{x}, \tau) - \mathbf{u}_t^{(B)}(\mathbf{x}, \tau) = [u_x(\mathbf{x}, \tau) \; u_y(\mathbf{x}, \tau)]^T$ is defined as the relative surface tangential displacement.

A detailed explanation and a schematic representation of the (relative) surface normal and tangential displacements ($\mathbf{u} = \left[\mathbf{u}_t(\mathbf{x}, \tau)^T \; u_n(\mathbf{x}, \tau)\right]^T$) at point $\mathbf{x}$ can be found in Kalker's book [84].

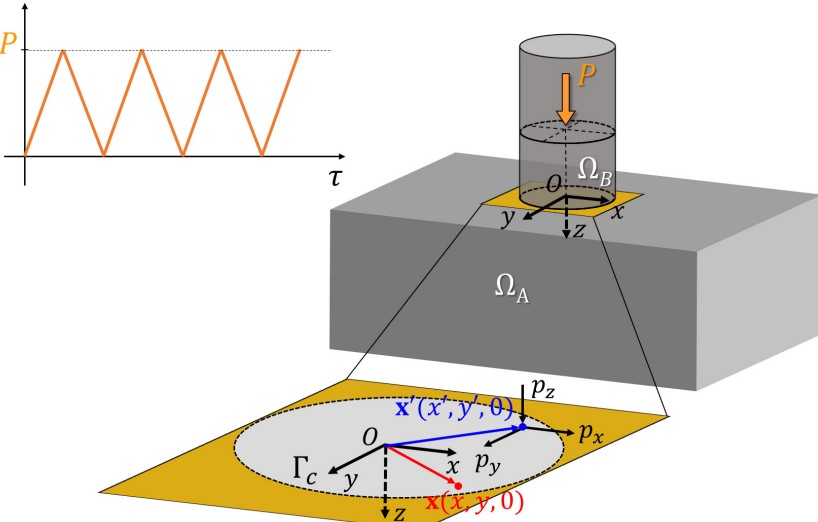

**Figure 1.** An elastic half-space ($\Omega_A$) and a cylindrical flat-ended punch ($\Omega_B$) come into contact under vertical loading and unloading cycles ($P$).

## 3. Boundary Element Approximation

The displacement difference $\mathbf{u}(\mathbf{x}, \tau)$ can be computed similarly to [77,85,86], which ignores the inertia effects (i.e., considers a quasi-static description). Under these assumptions,

the relation between the deformation $\mathbf{u} = [\mathbf{u}_t(\mathbf{x}, \tau)^T \; u_n(\mathbf{x}, \tau)]^T$ at point $\mathbf{x}$ and the contact traction $\mathbf{p} = [\mathbf{p}_t(\mathbf{x}, \tau)^T \; p_n(\mathbf{x}, \tau)]^T$ on points $\mathbf{x}' \in \Gamma_c$ can be expressed as

$$\mathbf{u}(\mathbf{x}, \tau) = \int \int_{\Gamma_c(\tau)} \mathbf{A}(\mathbf{x}, \mathbf{x}') \, \mathbf{p}(\mathbf{x}', \tau) \, dx' dy'. \tag{3}$$

In Equation (3), $\mathbf{A}(\mathbf{x}, \mathbf{x}')$ is the kernel function matrix, which quantifies the surface displacement components at $\mathbf{x}$ induced by one of the traction components of unit magnitude acting at $\mathbf{x}'$. It can be written as

$$\mathbf{A}(\mathbf{x}, \mathbf{x}') = \begin{bmatrix} A_{xx}(\mathbf{x}, \mathbf{x}') & A_{xy}(\mathbf{x}, \mathbf{x}') & A_{xz}(\mathbf{x}, \mathbf{x}') \\ A_{yx}(\mathbf{x}, \mathbf{x}') & A_{yy}(\mathbf{x}, \mathbf{x}') & A_{yz}(\mathbf{x}, \mathbf{x}') \\ A_{zx}(\mathbf{x}, \mathbf{x}') & A_{zy}(\mathbf{x}, \mathbf{x}') & A_{zz}(\mathbf{x}, \mathbf{x}') \end{bmatrix}, \tag{4}$$

where

$$
\begin{aligned}
&A_{xx}(\mathbf{x}, \mathbf{x}') = \frac{1}{\pi G}\left(\frac{1-\nu}{s} + \frac{\nu(x'-x)^2}{s^3}\right), \; A_{xy}(\mathbf{x}, \mathbf{x}') = \frac{\nu}{\pi G}\left(\frac{(x'-x)(y'-y)}{s^3}\right), \\
&A_{yy}(\mathbf{x}, \mathbf{x}') = \frac{1}{\pi G}\left(\frac{1-\nu}{s} + \frac{\nu(y'-y)^2}{s^3}\right), \; A_{yz}(\mathbf{x}, \mathbf{x}') = \frac{K}{\pi G}\left(\frac{y'-y}{s^2}\right), \\
&A_{zz}(\mathbf{x}, \mathbf{x}') = \frac{1}{\pi G}\left(\frac{1-\nu}{s}\right), \; A_{xz}(\mathbf{x}, \mathbf{x}') = \frac{K}{\pi G}\left(\frac{x'-x}{s^2}\right), \\
&A_{yx}(\mathbf{x}, \mathbf{x}') = A_{xy}(\mathbf{x}, \mathbf{x}'), \; A_{zx}(\mathbf{x}, \mathbf{x}') = -A_{xz}(\mathbf{x}, \mathbf{x}'), \; A_{zy}(\mathbf{x}, \mathbf{x}') = -A_{yz}(\mathbf{x}, \mathbf{x}'),
\end{aligned} \tag{5}
$$

where $s = \sqrt{(x'-x)^2 + (y'-y)^2}$ and $G$, $\nu$, and $K$ are the material parameters, defined as

$$\frac{1}{G} = \frac{1}{2}\left(\frac{1}{G^{(A)}} + \frac{1}{G^{(B)}}\right), \; \frac{\nu}{G} = \frac{1}{2}\left(\frac{\nu^{(A)}}{G^{(A)}} + \frac{\nu^{(B)}}{G^{(B)}}\right), \; K = \frac{G}{4}\left(\frac{1-2\nu^{(A)}}{G^{(A)}} - \frac{1-2\nu^{(B)}}{G^{(B)}}\right). \tag{6}$$

The terms of Equation (4) can also be expressed as: $\mathbf{A}(\mathbf{x}, \mathbf{x}') = \mathbf{A}(\mathbf{x}' - \mathbf{x})$, i.e., indicating that the influence coefficients depend on the relative positions of the two surface points $\mathbf{x}$ and $\mathbf{x}'$.

## 4. Contact and Wear Laws

### 4.1. Normal Contact Restrictions

Signorini's unilateral contact conditions for the normal gap $g_n(\mathbf{x}, \tau)$ and the normal contact pressure $p_n = p_z(\mathbf{x}, \tau)$ can be expressed, according to Alart and Curnier [87], as

$$p_n - \mathbb{P}_{\mathbb{R}^+}(p_n^*) = 0, \tag{7}$$

where $\mathbb{P}_{\mathbb{R}^+}(\bullet) = max(0, \bullet)$ is the normal projection function and $p_n^* = p_n + r_n g_n$ is the augmented normal traction—$r_n$ being the normal penalty parameter ($r_n \in \mathbb{R}^+$).

### 4.2. Tangential Contact Restrictions

Similarly, the frictional contact constraints can also be formulated, according to [88], as

$$\mathbf{p}_t - \mathbb{P}_{\mathbb{E}_\rho}(\mathbf{p}_t^*) = 0, \tag{8}$$

where $\mathbf{p}_t^* = \mathbf{p}_t - r_t \mathbb{M}^2 \dot{\mathbf{g}}_t$ is the augmented tangential traction (where $\mathbb{M} = diag(\mu_1, \mu_2)$ and $r_t \in \mathbb{R}^+$) and $\mathbb{P}_{\mathbb{E}_\rho}(\bullet) : \mathbb{R}^2 \longrightarrow \mathbb{R}^2$ is the tangential projection function:

$$\mathbb{P}_{\mathbb{E}_\rho}(\mathbf{p}_t^*) = \begin{cases} \mathbf{p}_t^* & \text{if } ||\mathbf{p}_t^*||_\mu < \rho, \\ \rho \, \mathbf{p}_t^* / ||\mathbf{p}_t^*||_\mu & \text{if } ||\mathbf{p}_t^*||_\mu \geq \rho. \end{cases} \tag{9}$$

In the expression above, $\rho = |\mathbb{P}_{\mathbb{R}^+}(p_n^*)|$ and the elliptic norm $|| \bullet ||_\mu$ is defined, so that

$$||\mathbf{p}_t||_\mu = \sqrt{(p_{e_1}/\mu_1)^2 + (p_{e_2}/\mu_2)^2}, \tag{10}$$

where $\mu_1$ and $\mu_2$ are the principal friction coefficients in the directions $\{e_1, e_2\}$. Moreover, the tangential contact tractions' components—and the tangential slip velocity components— can be expressed in the tribological axes $\{e_1, e_2\}$ as

$$\begin{bmatrix} p_{e_1} \\ p_{e_2} \end{bmatrix} = \begin{bmatrix} \cos\beta & \sin\beta \\ -\sin\beta & \cos\beta \end{bmatrix} \begin{bmatrix} p_x \\ p_y \end{bmatrix}, \tag{11}$$

where the angle $\beta$ is defined in [71] as the tribological axes' angle orientation. In the expression above, the tangential slip velocity $\dot{\mathbf{g}}_t$ can be expressed at time $\tau_k$ as follows: $\dot{\mathbf{g}}_t \approx \Delta\mathbf{g}_t/\Delta\tau$ (see [88–90]), where $\Delta\mathbf{g}_t = \mathbf{g}_t(\tau_k) - \mathbf{g}_t(\tau_{k-1})$ and $\Delta\tau = \tau_k - \tau_{k-1}$.

*4.3. Wear Law*

This work assumed a Holm–Archard wear law [91], which, for an infinitesimally small apparent contact area, can be expressed in terms of the wear rate [33,43,47,49,50,92–95], as: $\dot{w} = i_w |p_n| ||\dot{\mathbf{g}}_t||$, $i_w$ being the wear coefficient.

Since orthotropic tribological properties were considered in this work, an orthotropic wear law [50,96,97] should be assumed. Therefore, the wear law should be rewritten as

$$\dot{w} = |p_n| ||\dot{\mathbf{g}}_t||_i, \tag{12}$$

where

$$||\dot{\mathbf{g}}_t||_i = \sqrt{(i_1 \dot{g}_{e_1})^2 + (i_2 \dot{g}_{e_2})^2}, \tag{13}$$

where $i_1$ and $i_2$ are the principal intensity wear coefficients.

The derivatives can be expressed—under quasi-static wear and contact conditions as $\dot{w} \simeq \Delta w = w(\tau_k) - w(\tau_{k-1})$ and $\dot{\mathbf{g}}_t \simeq \Delta\mathbf{g}_t = \mathbf{g}_t(\tau_k) - \mathbf{g}_t(\tau_{k-1})$. Therefore, the wear depth at the instant $\tau_k$ can be computed as

$$w(\tau_k) = w(\tau_{k-1}) + p_n(\tau_k) || \mathbf{g}_t(\tau_k) - \mathbf{g}_t(\tau_{k-1})||_i. \tag{14}$$

**5. Subsurface Stresses**

Finally, the subsurface stresses at point $\mathbf{x} \in \Omega^{(\alpha)}$ ($\alpha = A, B$)—caused by a surface contact pressure at $\mathbf{x}' \in \Gamma_c$ (see Figure 2a)—can be computed as

$$\sigma(\mathbf{x}, \tau) = \int\int_{\Gamma_c(\tau)} \mathbf{T}(\mathbf{x}, \mathbf{x}') \, \mathbf{p}(\mathbf{x}', \tau) \, dx'dy', \tag{15}$$

where the influence coefficients of the kernel function $\mathbf{T}(\mathbf{x}, \mathbf{x}')$ depend on the relative positions of the two surface points $\mathbf{x}$ and $\mathbf{x}'$; therefore, $\mathbf{T}(\mathbf{x}, \mathbf{x}') = \mathbf{T}(\mathbf{x}' - \mathbf{x})$.

For the sake of clarity, the expression above is rewritten as

$$\sigma_{ij}(\mathbf{x}, \tau) = \int\int_{\Gamma_c(\tau)} T_{ij}^{Sx}(\mathbf{x}' - \mathbf{x}) \, p_x(\mathbf{x}', \tau) + T_{ij}^{Sy}(\mathbf{x}' - \mathbf{x}) \, p_y(\mathbf{x}', \tau) +$$

$$T_{ij}^N(\mathbf{x}' - \mathbf{x}) \, p_n(\mathbf{x}', \tau) \, dx'dy'. \tag{16}$$

The explicit expressions of $T_{ij}^N(x, y, z)$, $T_{ij}^{Sx}(x, y, z)$ and $T_{ij}^{Sy}(x, y, z)$ can be found in [71].

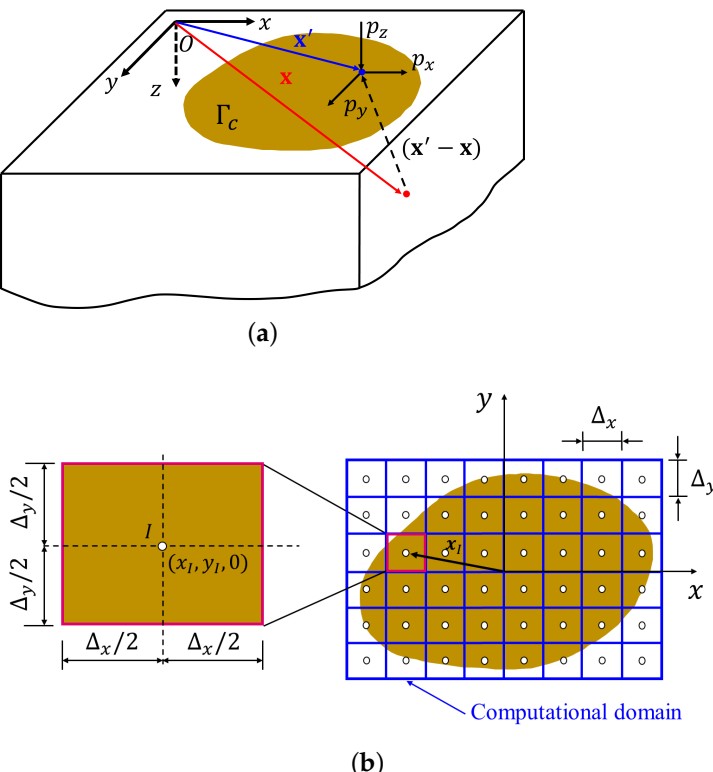

(a)

(b)

**Figure 2.** (**a**) Half-space scheme where surface tractions on a point $\mathbf{x}' \in \Gamma_c$ cause subsurface stresses at point $\mathbf{x} \in \Omega^{(\alpha)}$ ($\alpha = A, B$). (**b**) A mesh scheme considered for the potential contact zone.

## 6. Solution Scheme

The nonlinear equations' set Equations (1)–(3), (7), (8), (14) and (16) was discretized and solved by using the numerical scheme presented by the authors in [71]. This allowed us to compute the contact conditions and the subsurface stresses' evolution caused by the contact—under wear conditions—on every loading and unloading cycle (*k*). For the numerical simulation, a rectangular potential contact zone was considered and discretized by a regular mesh with $N_e = N_x \times N_y$ elements of size $\Delta_x \times \Delta_y$ (see Figure 2b), whose coordinates were the center of each element. Subsurface stresses for a set of interior points can be computed from the discrete expression of Equation (16).

## 7. Numerical Analysis

This work analyzed the influence of friction and wear on the surface and subsurface contact stresses generated—in an elastic half-space—by the cylindrical flat-ended punch, under normal cyclic indentation loading. Moreover, orthotropic tribological laws were also considered in the analysis. For this purpose, we considered a cylindrical punch, whose radius was $a_o = 1.8$ mm. Both domains were constituted by linear and elastic materials with the same Poisson's ratios $\nu^{(A)} = \nu^{(B)} = 0.3$ and different Young's modulus, i.e., $E^{(B)}/E^{(A)} = 100$, where $E^{(A)} = 200 \times 10^3$ MPa. Therefore, the punch can be considered as rigid compared to the half-space.

This section is divided into three blocks. Firstly, in Section 7.1, we validate the computational framework by solving a benchmark frictionless indentation problem—under static normal load (*P*)—whose theoretical surface contact tractions and subsurface stress distributions can be found in the literature [76,98]. In this context, the influence of friction was also studied. Therefore, considering several values of the friction coefficient ($\mu$), we can study its influence on the contact tractions and the subsurface stresses. Secondly, in Section 7.2, the cylindrical flat-punch is subjected to a cyclic normal load—loading and unloading cycles—which induces wear on the solids' surfaces, i.e., fretting wear conditions.

Therefore, this section studies the influence of both friction and wear on the evolution of the surface contact tractions and subsurface contact stresses, assuming isotropic friction and wear laws. Finally, Section 7.3 studies the cylindrical flat-punch under cyclic normal load conditions and orthotropic tribological laws.

### 7.1. Cylindrical Flat-Punch under Static Normal Load Conditions

First, we validated the accuracy of the proposed formulation to solve this problem by solving the static indentation under frictionless contact conditions ($\mu = 0$). The theoretical solution of this benchmark problem can be found in [76] or [98]. For the normal contact pressure, it can be presented as a function of the radial distance ($r = \sqrt{x^2 + y^2}$) as

$$p_n = \frac{P}{2\pi a_o \sqrt{a_o^2 - r^2}} \quad (r \leq a_o), \tag{17}$$

where $P$ is the centrally applied vertical force. This resulting normal load ($P$) and the average contact pressure ($p_o$) in the contact zone can be written, respectively, as a function of the normal indentation ($g_o$) as

$$P = 2E^* a_o g_{n,o}, \tag{18}$$

and

$$p_o = \frac{P}{\pi a_o^2} = \frac{2E^* g_{n,o}}{\pi a_o}, \tag{19}$$

where $E^* = \left( (1 - \nu^{(A)\,2})/E^{(A)} + (1 - \nu^{(B)\,2})/E^{(B)} \right)^{-1}$.

A comparison between the theoretical normal contact pressure distribution, for $P = 714$ N ($g_{n,o} = 9113.4$ µm), and the numerical results is presented in Figure 3a. In this figure, all the variables are presented in a non-dimensional form. The $x$-axis coordinates are expressed relative to $a_o$, and the normal contact pressure is presented relative to $p_o$. This value can be related to the average pressure in the contact ($p_o$). An excellent agreement between the numerical results—square markers—and the theoretical solution—continuous line—can be observed in Figure 3a, the maximum values of the normal contact pressure ($p_{n,max}$) being located at the contact zone limits, i.e., at $x/a_o \approx 1.0$.

Now, we can study the influence of friction on the surface contact tractions.

The normal contact of a rigid flat-ended cylindrical punch with an elastic half-space under complete stick conditions was initially solved by Mossakovskii [99]. This work provided us the contact stiffness ($k_n := P/g_{n,o}$), i.e., the ratio between the load on the punch ($P$) and the indentation ($g_{n,o}$), for the mentioned tangential stick conditions. Therefore, this contact stiffness can be related to the contact stiffness for the frictionless case, presented in Equation (18). The resulting relation between the contact stiffness for the complete stick and for frictionless contact can be expressed as

$$k_n|_{adhesion}/k_n|_{\text{frictionless}} = (1 - \nu) \ln (3 - 4\nu)/(1 - 2\nu). \tag{20}$$

Figure 3b shows the relation between the computed contact stiffness for frictional and for frictionless contact ($k_n|_\mu/k_n|_{\text{frictionless}}$) as a function of the friction coefficient ($\mu$) and the material parameter $\nu$. Moreover, the relation between the contact stiffness for the complete stick and for frictionless contact ($k_n|_{adhesion}/k_n|_{\text{frictionless}}$)—presented by Mossakovskii [99]—is also included. We can observe how the computed $k_n|_\mu$ tends to the contact stiffness for the complete stick case ($k_n|_{adhesion}$) when high values for the friction coefficient are considered. In particular, for the material properties considered ($\nu = 0.3$), the contact stiffness value reached the adhesion case values ($k_n|_\mu \approx k_n|_{adhesion}$) when $\mu \geq 0.4$.

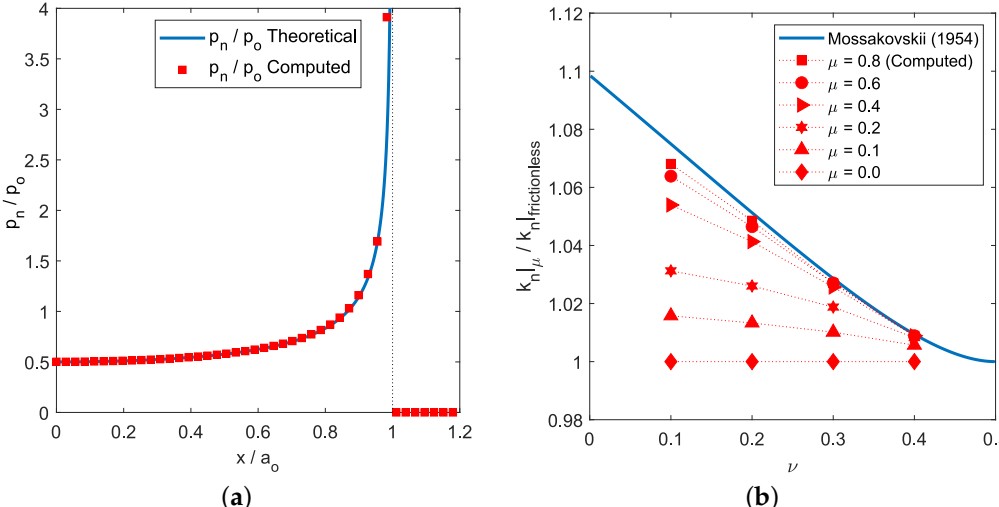

**Figure 3.** (**a**) Theoretical—continuous blue line—and numerical—square markers—results of the comparison between the theoretical normal contact pressure distribution—continuous line—and the numerical—square markers—for a frictionless contact indentation. (**b**) Relation between the computed contact stiffness for frictional and for frictionless contact ($k_n|_\mu/k_n|_\text{frictionless}$) as a function of the friction coefficient ($\mu$) and the material parameter ($\nu$). Moreover, the relation between the contact stiffness for the complete stick and for frictionless contact ($k_n|_\text{adhesion}/k_n|_\text{frictionless}$)—presented by Mossakovskii [99]—is also included.

In Figure 4, we present the normal and tangential contact tractions' distributions under isotropic frictional contact conditions, i.e., $\mu_1 = \mu_2 = \mu = \{0.1, 0.15, 0.2, 0.25, 0.3, 0.4\}$. The results showed how the stick (or adhesion) region in the contact zone (i.e., where $|p_x| < \mu p_o$), increased with the value of the friction coefficient ($\mu$) until a threshold value is reached (i.e., $\mu \approx 0.4$ in our analysis). Then, all the points of the contact area reached a contact–stick state, and additional increments on $\mu$ did not cause an increment on the tangential contact tractions.

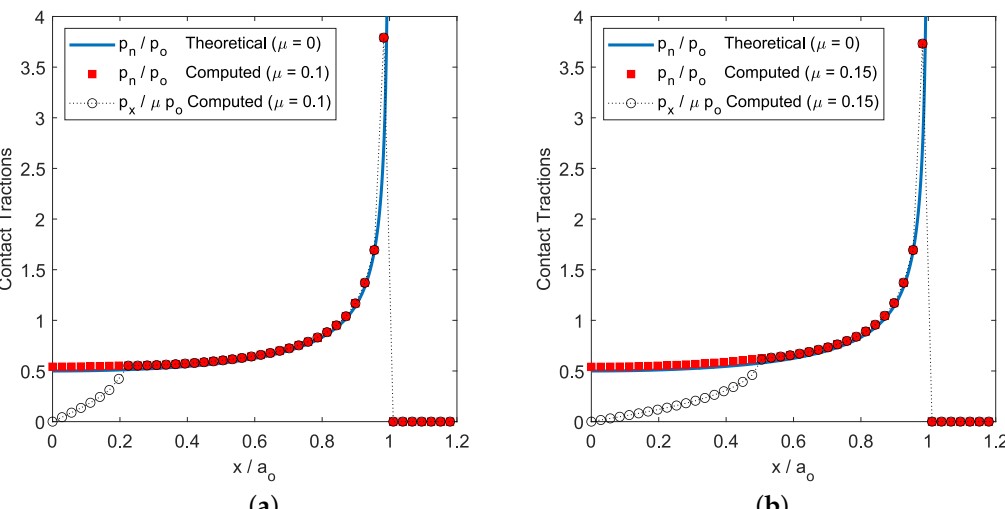

**Figure 4.** *Cont.*

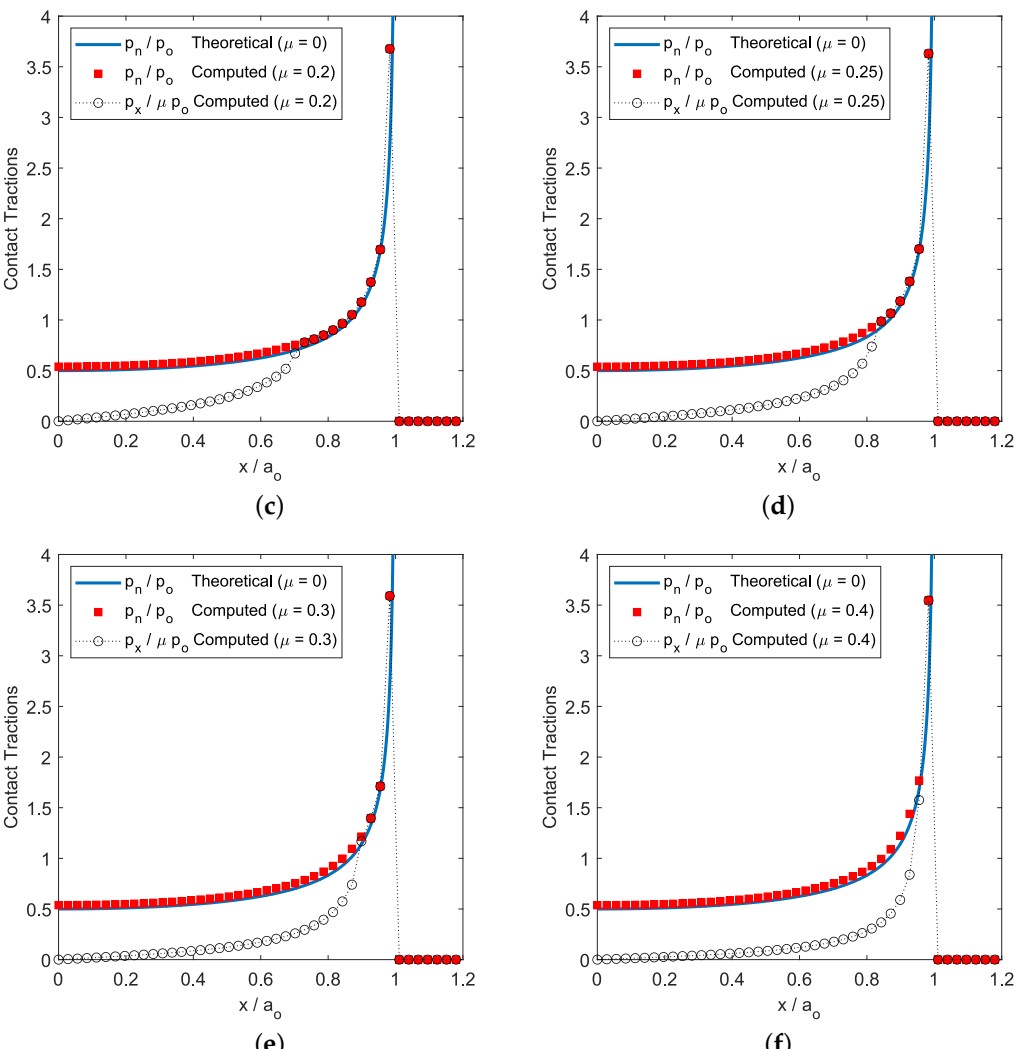

**Figure 4.** Normal and tangential contact pressure distributions under isotropic frictional contact conditions: $\mu_1 = \mu_2 = \mu = \{0.1, 0.15, 0.2, 0.25, 0.3, 0.4\}$ (**a**–**f**), respectively.

To validate the subsurface stress components' computational framework, Figure 5a,b presents, respectively, the theoretical and numerical von Mises equivalent stress distributions in the *x*-*z*-plane of the elastic half-space, normalized by $p_o$. We can observe an excellent agreement between the theoretical von Mises stress distribution—presented in Figure 5a—and the numerical solution for the frictionless case: $\mu_1 = \mu_2 = \mu = 0$; see Figure 5b. The theoretical von Mises equivalent stress presented in Figure 5a can be obtained from

$$\sigma_{VM} = \sqrt{\frac{1}{2}\left((\sigma_{rr} - \sigma_{\phi\phi})^2 + (\sigma_{\phi\phi} - \sigma_{zz})^2 + (\sigma_{zz} - \sigma_{rr})^2 + 6\tau_{rz}^2\right)}, \tag{21}$$

where the expressions of the cylindrical stress components are presented in Appendix A. We can see in Figure 5a how a stress singularity exists at the edge of the contact circle ($r = a_o$).

It should be noted that we considered the von Mises failure criterion since ductile contacting bodies—under elastic half-space assumptions—were considered in the analyses. This criterion would not be valid for hardened surfaces or hard wear-resistant coatings such as ceramics. For layered surfaces, the readers should refer to the work of Teodorescu et al. [60].

Once the subsurface stresses were validated, we studied the influence of friction on the subsurface von Mises stress distribution in Figure 5b–f. For the frictionless case,

the maximum values of the normalized von Mises stress distribution ($\sigma_{VM}/p_o$) were located at $x/a_o = 1.0$ and $z/a_o = 0.0$. We also can distinguish a subsurface region, where $\sigma_{VM}/p_o \geq 1$ (marked with black lines in the figures). Moreover, when we increased the friction intensity, i.e., $\mu_1 = \mu_2 = \mu = \{0.1, 0.2, 0.3, 0.4\}$, the normalized von Mises stress distributions presented in Figure 5c–f increased the size of the region with $\sigma_{VM}/p_o \geq 1$ and, additionally, created a new stress gradient around the $z$-axis. However, when the friction intensity reached the mentioned threshold value, no more changes were observed in the normalized von Mises stress distribution.

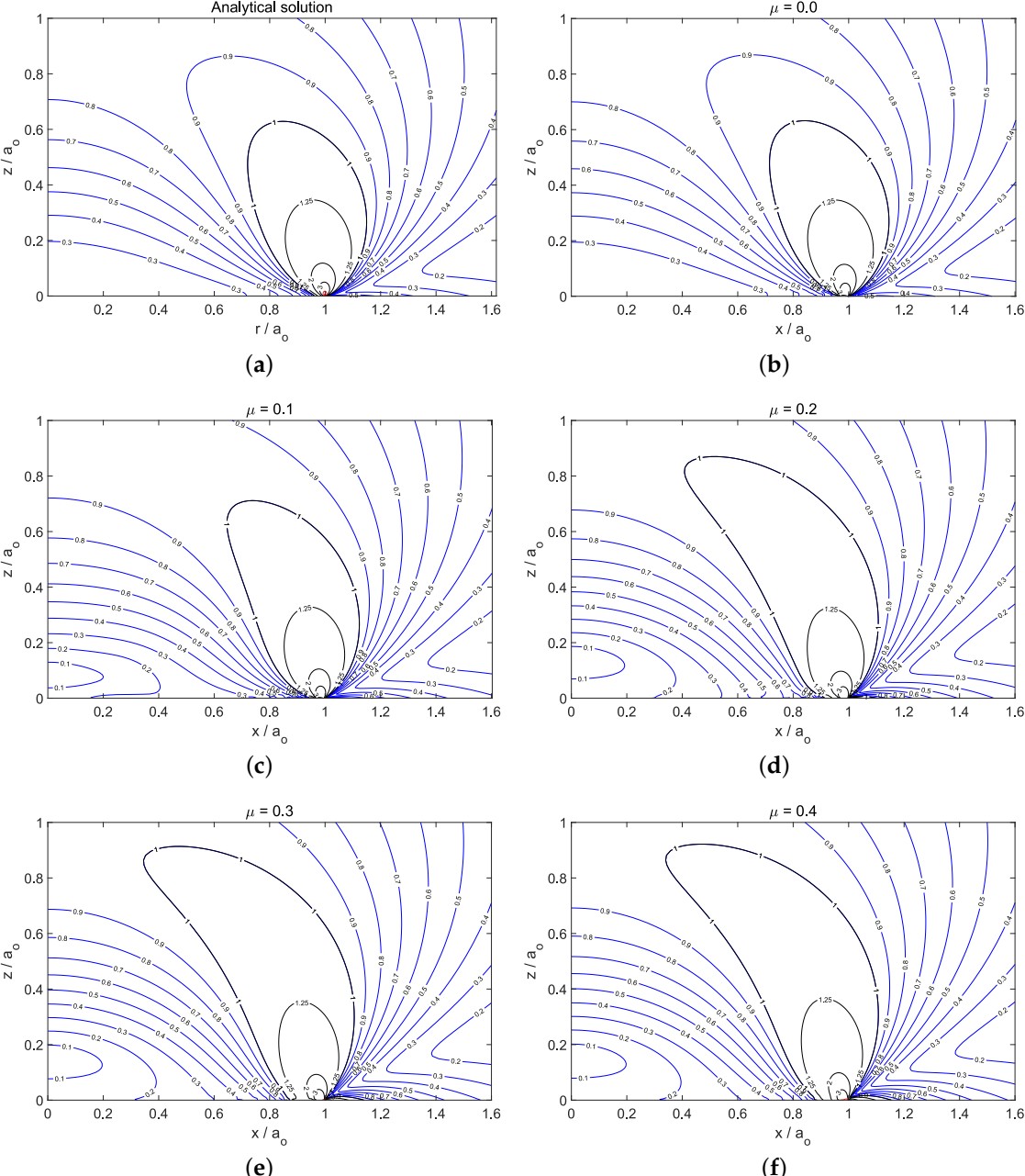

**Figure 5.** (**a**) Theoretical von Mises equivalent stress distribution in the half-space—caused by a cylindrical flat-punch indentation—which is normalized by the average pressure ($p_o$). (**b**) Normalized—computed—von Mises equivalent stress in the half-space caused by a cylindrical flat-punch indentation ($\mu_1 = \mu_2 = \mu = 0$). (**c**–**f**) Normalized von Mises equivalent stress distributions computed in the elastic half-space under isotropic frictional contact conditions: $\mu_1 = \mu_2 = \mu = \{0.1, 0.2, 0.3, 0.4\}$, respectively.

### 7.2. Cyclic Normal Load Conditions and Isotropic Tribological Laws

This section studies the influence of the friction intensity on the evolution of the surface tractions and subsurface stresses when the solids are subjected to—isotropic—fretting wear conditions. In this case, the cylindrical flat-ended punch was subjected to normal indentation loading and unloading cycles over the elastic half-space (see Figure 1), the maximum load value being $P = 750$ N. The analyses were developed considering different values of the friction coefficients, i.e., $\mu_1 = \mu_2 = \mu$ (where $\mu = \{0.1, 0.2, 0.4\}$), and the wear coefficients values: $i_1 = i_2 = i_\omega$ (where $i_\omega = 1.33 \times 10^{-7}$ MPa$^{-1}$).

The influence of the friction intensity is clearly observed in Figure 6. We can see in Figure 6a the computed maximum wear depth evolution as a function of the number of load cycles (N), for several values of the friction coefficient: $\mu = \{0.1, 0.2, 0.4\}$. Figure 6b shows the wear volume evolution with the number of load cycles and the same values of the friction coefficient. In both figures, we can observe how the smaller the friction coefficient value was, the greater the wear damage (i.e., maximum wear depth or wear volume) obtained. This was due to the fact that the smaller the friction coefficient value was, the bigger the annular sliding contact zone obtained (see Figure 4a,f).

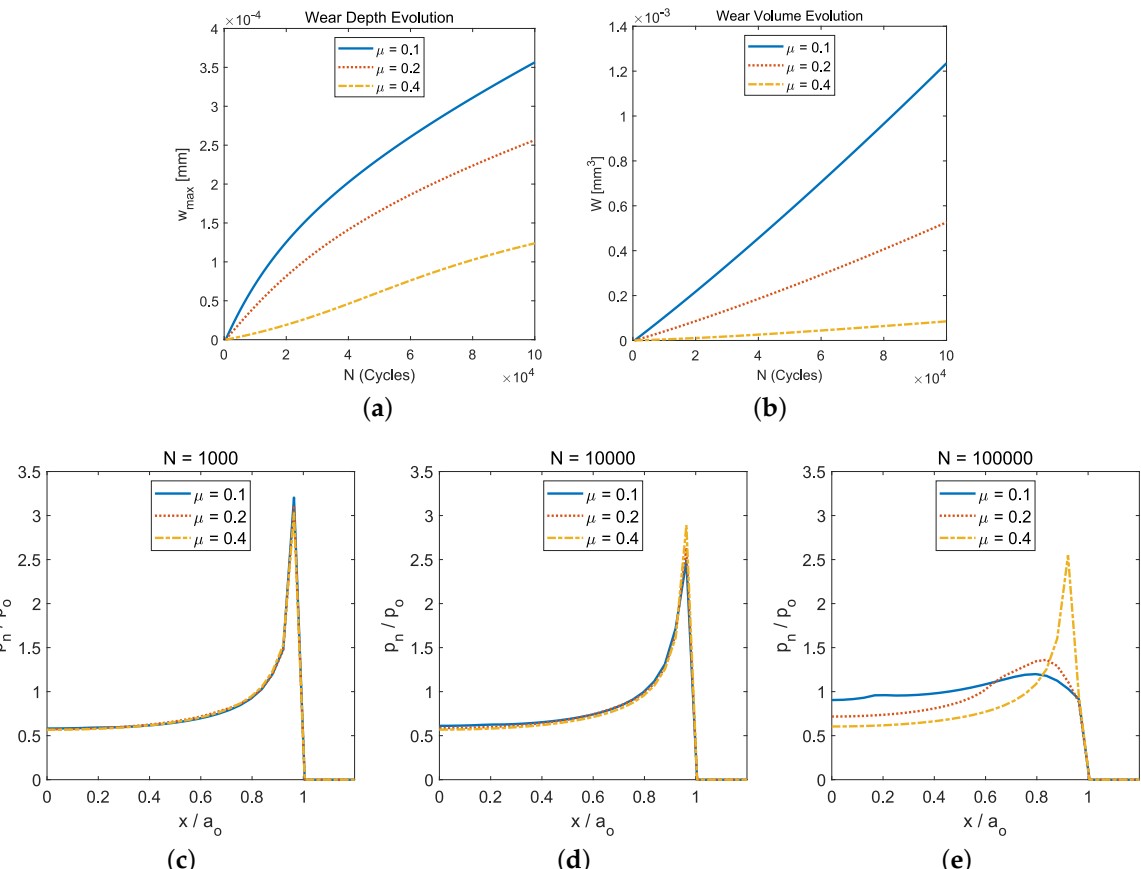

**Figure 6.** (**a**) Computed maximum wear depth evolution as a function of the number of load cycles (N), for several values of the friction coefficient: $\mu = \{0.1, 0.2, 0.4\}$. (**b**) Wear volume evolution with the number of load cycles and different values of the friction coefficient. (**c**–**e**) Normal contact pressure distributions after $10^3$, $10^4$, and $10^5$ load cycles, respectively, and several friction values of the friction coefficient: $\mu = \{0.1, 0.2, 0.4\}$.

The influence of the friction intensity in the evolution of the normal contact pressure distribution is presented in Figure 6c–e. They present the normal contact pressure distributions after $10^3$, $10^4$, and $10^5$ load cycles, respectively, and several friction values of the friction coefficient: $\mu = \{0.1, 0.2, 0.4\}$. We can observe in Figure 6e the tremendous differ-

ence between the normal contact pressure distribution obtained—after $10^5$ load cycles—for $\mu = 0.1$ and $\mu = 0.4$, respectively.

These differences can also be observed in Figure 7a–f for the computed wear depth distributions and normal contact pressure distribution, respectively. These surface distributions are presented in the $x$-$y$-plane, after $N = 10^5$ cycles, for the friction coefficient values: $\mu = \{0.1, 0.2, 0.4\}$. Due to the isotropic friction and wear laws considered, the distributions presented a $z$-axis symmetry.

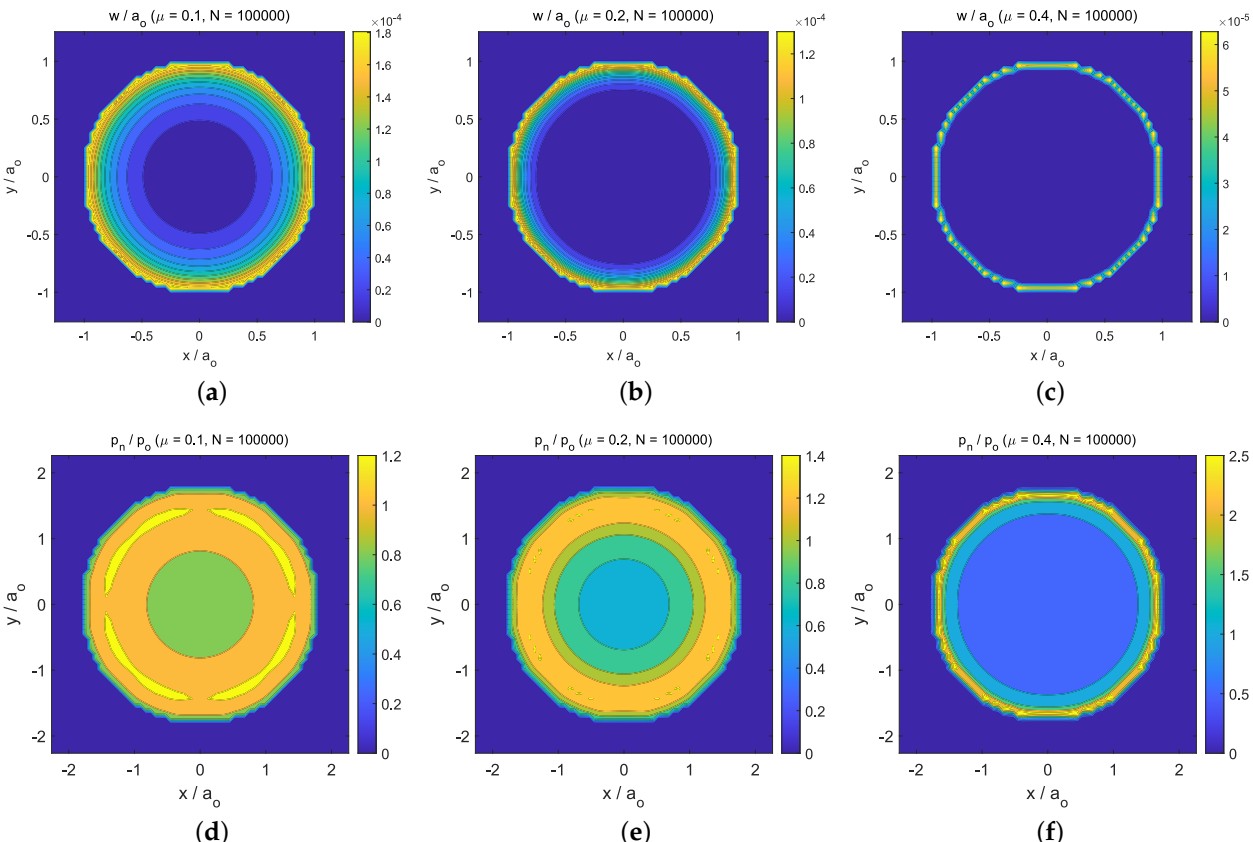

**Figure 7.** (**a**–**c**) Computed wear depth distribution after $N = 10^5$ cycles, for several values of the friction coefficient: $\mu = \{0.1, 0.2, 0.4\}$, respectively. (**d**–**f**) Resulting normal contact pressure distribution after $N = 10^5$ cycles, for $\mu = \{0.1, 0.2, 0.4\}$, respectively.

Finally, Figure 8 presents the subsurface von Mises stress distributions in the half-space after $N = 10^3$ cycles (see Figure 8a) and $N = 10^5$ cycles (see Figure 8b), for $\mu = \{0.1, 0.2, 0.4\}$, respectively. The stress distributions are presented in the $x$-$z$-plane, all the variables being non-dimensional. All the subsurface stress distributions obtained for $\mu = \{0.1, 0.2, 0.4\}$ after a low number of cycles ($N = 10^3$)—see Figure 8a—presented a maximum stress value at the edge of the contact circle, its radius being slightly reduced from $a_o$ due to wear. However, after a big number of load cycles (i.e., $N = 10^5$), the friction intensity had a big influence not only on the resulting values of the von Mises stress, but also on the location of the maximum stress value—see Figure 8b. On the one hand, since low friction values (i.e., $\mu = 0.1$) caused important wear damage on the solid surfaces, this also led to a stress distribution where the maximum value of the subsurface von Mises stress was located at $r = 0$ and $z/a_o \approx 0.7$. On the other hand, when high friction values were considered (i.e., $\mu = 0.4$), low wear damage was produced on the solid surfaces, and therefore, the stress distribution presented its maximum value located on the surface $z/a_o \approx 0$ and at the edge of the contact circle ($r/a_o \approx 0.9$).

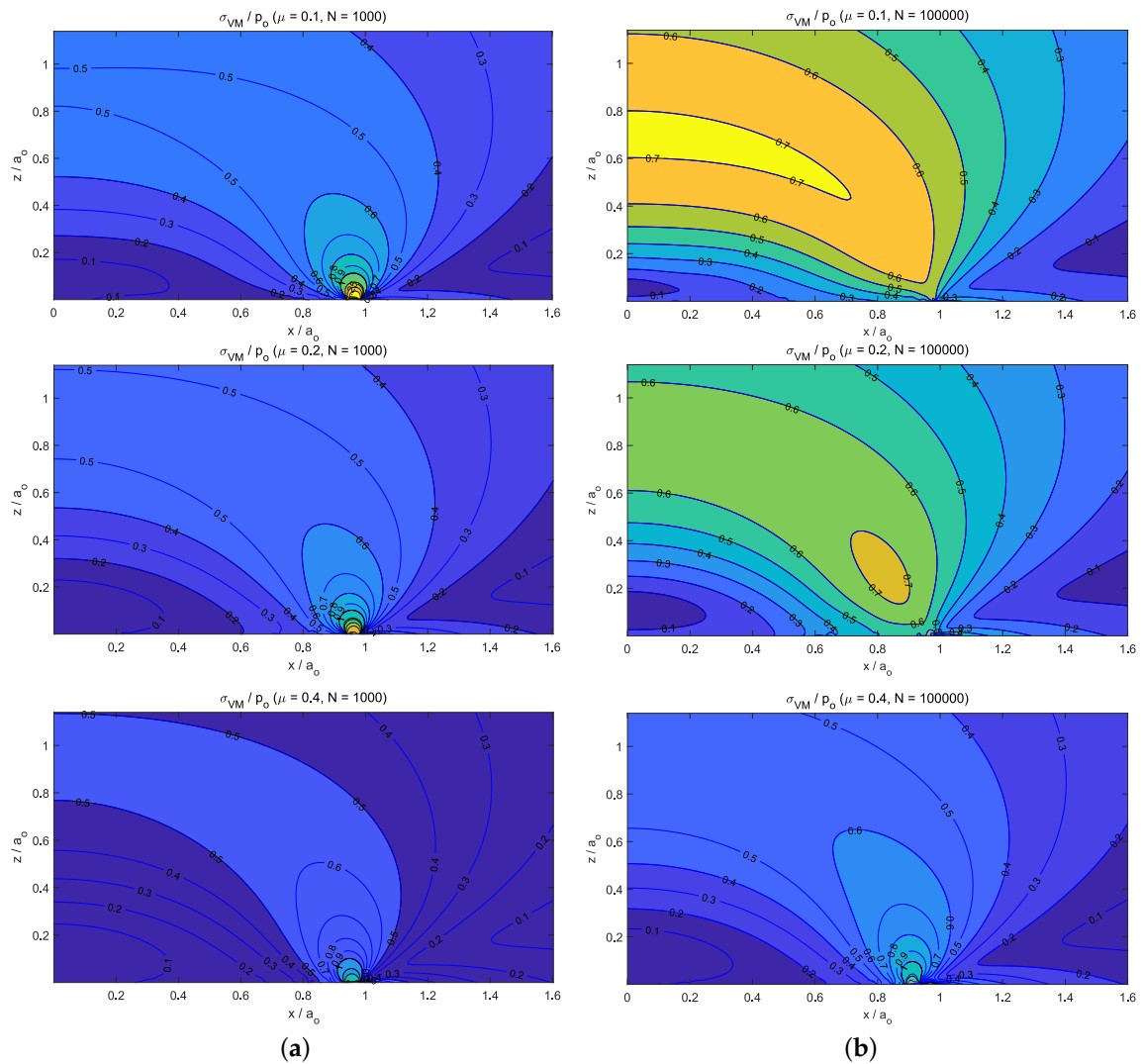

**Figure 8.** von Mises equivalent stress distributions in the half-space after: (**a**) $N = 10^3$ cycles and (**b**) $N = 10^5$ cycles, for $\mu = \{0.1, 0.2, 0.4\}$, respectively. The stress distributions are presented in the $x$-$z$-plane.

### 7.3. Cyclic Normal Load Conditions and Orthotropic Tribological Laws

This section studies the influence of orthotropic friction and wear laws (i.e, under $\mu_1 \neq \mu_2$ and $i_1 \neq i_2$) on the evolution of the surface tractions and the subsurface stresses when fretting wear conditions are considered. Similar to the previous section, the cylindrical flat-ended punch was subjected to normal indentation loading and unloading cycles over the elastic half-space, the maximum load value being $P = 750$ N. However, in this case, we considered the following values for the friction coefficients: $\mu_1 = 0.1$ and $\mu_2 = 0.4$, and the wear coefficients: $i_1 = 1.33 \times 10^{-7}$ MPa$^{-1}$ and $i_2 = i_1 \mu_2 / \mu_1$ (in order to maintain that the wear rate is proportional to the friction dissipation energy).

To validate this study, we present in Figure 9a,b the computed maximum wear depth evolution and the total wear volume, respectively, as a function of the number of load cycles (N), for several values of the orientation of the tribological axes' angles: $\beta = \{0°, 45°, 90°\}$. In both figures, we can observe how these magnitudes were invariant with the angle $\beta$. Additionally, it can be observed that the maximum wear depth evolution for the orthotropic case was similar to the average evolution between the isotropic fretting wear cases presented in Figure 6 for $\mu = 0.1$ and $\mu = 0.4$. However, the wear volume evolution did not follow that pattern.

Figure 10a–c present the computed wear depth distribution after $N = 10^5$ cycles, for the following values of the tribological axes' angle orientation: $\beta = \{0°, 45°, 90°\}$,

respectively. The resulting normal contact pressure distributions—after $N = 10^5$ cycles—are presented in Figure 10d–f, for $\beta = \{0°, 45°, 90°\}$, respectively. Contrary to what Figure 7 showed for the wear depth and the normal contact pressure distributions under isotropic fretting wear conditions, the distributions in Figure 10 for orthotropic fretting wear conditions did not present $z$-axis symmetry. The maximum value of the normal contact pressure distributions was located where the wear depth distributions presented the greatest gradient values. This region was located at the intersection of the edge of the contact circle and the direction of the tribological principal axis with the greatest friction coefficient (i.e., $e_2$-axis in this case, since $\mu_2 > \mu_1$).

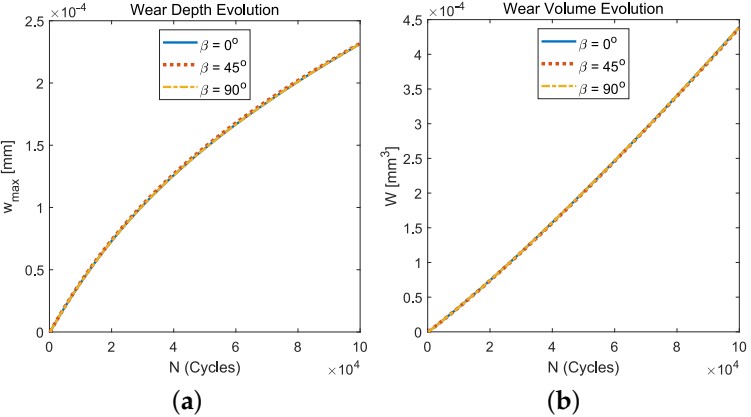

(a)  (b)

**Figure 9.** (**a**) Computed maximum wear depth evolution as a function of the number of load cycles (N), for the following values of the tribological axes' angle orientation: $\beta = \{0°, 45°, 90°\}$. (**b**) Wear volume evolution with the number of load cycles and different values of $\beta$.

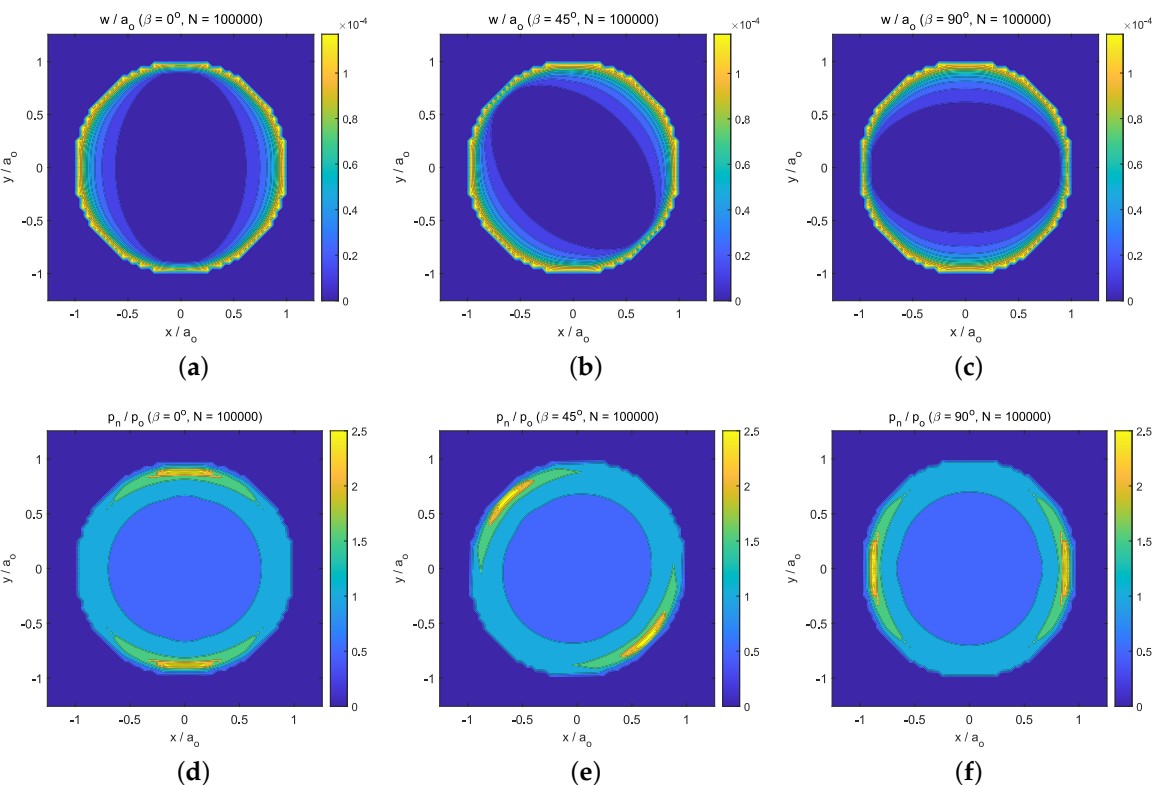

(a)  (b)  (c)

(d)  (e)  (f)

**Figure 10.** (**a**–**c**) Computed wear depth distribution after $N = 10^5$ cycles, for the following values of the tribological axes' angle orientation: $\beta = \{0°, 45°, 90°\}$, respectively. (**d**–**f**) Resulting normal contact pressure distribution after $N = 10^5$ cycles, for $\beta = \{0°, 45°, 90°\}$, respectively.

Finally, Figure 11 presents the subsurface von Mises stress distributions in the half-space after $N = 10^3$ cycles (see Figure 11a) and $N = 10^5$ cycles (see Figure 11b), for $\beta = \{0°, 45°, 90°\}$, respectively. Again, the stress distributions are presented in the *x-z*-plane, all the variables being non-dimensional: the *x* and *z* coordinates are expressed relative to $a_o$, and the von Mises equivalent stress is presented relative to $p_o$. All the subsurface stress distributions obtained for $\beta = \{0°, 45°, 90°\}$ after a low number of cycles ($N = 10^3$ cycles)—see Figure 11a—presented a stress maximum value at the edge of the contact circle, which, due to wear, was slightly less than $a_o$. The same location of the maximum stress value (i.e., the subsurface region in the edge of the contact circle) was observed after $N = 10^5$ cycles—see Figure 11b. These discrepancies between the stress distributions obtained under isotropic and orthotropic fretting wear conditions can be explained due to the lack of *z*-axis symmetry in the wear depth distribution obtained under orthotropic fretting wear conditions.

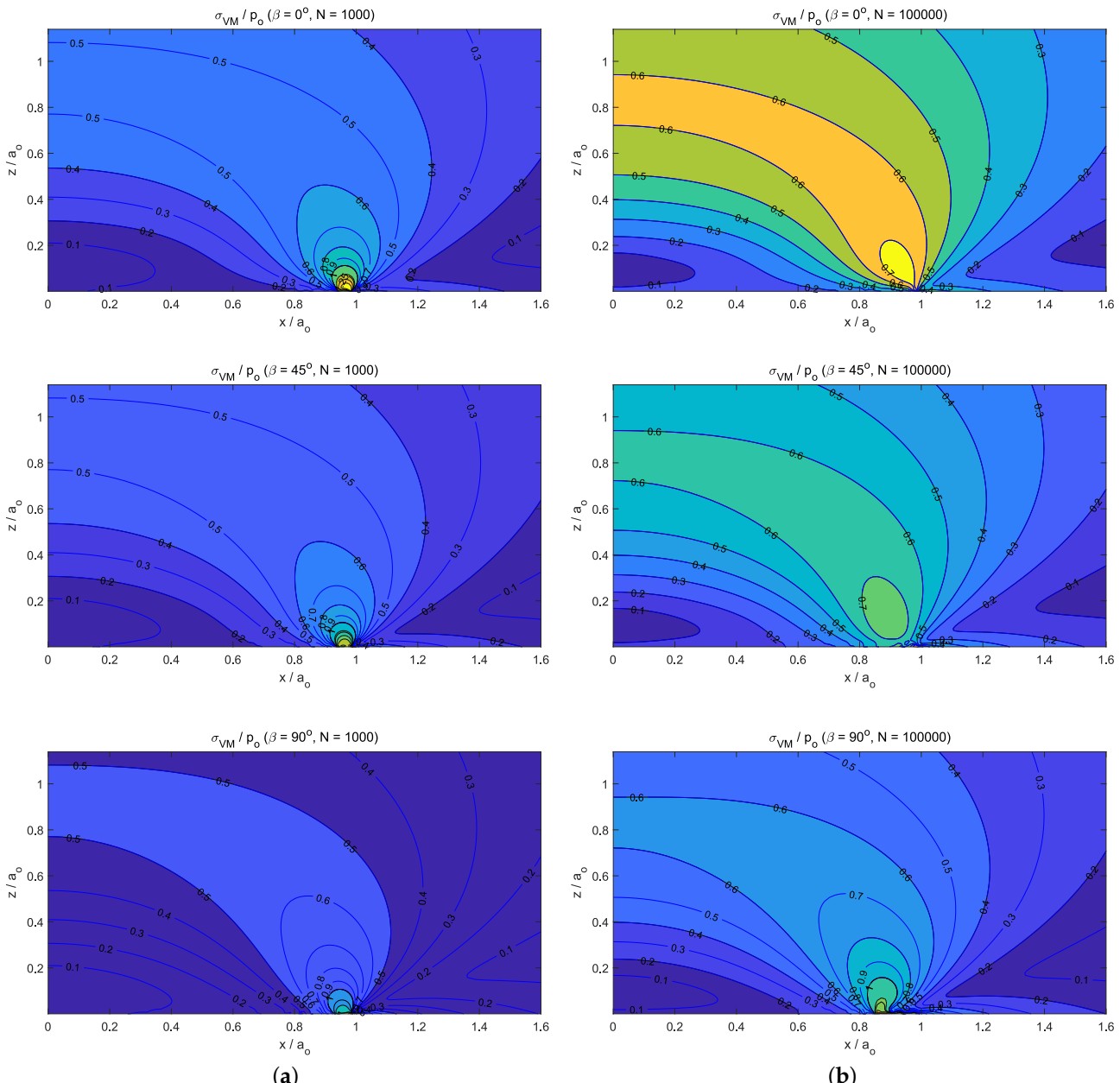

**Figure 11.** von Mises equivalent stress distributions in the half-space after: (**a**) $N = 10^3$ cycles and (**b**) $N = 10^5$ cycles, for $\beta = \{0°, 45°, 90°\}$, respectively.

## 8. Summary and Conclusions

This work analyzed the influence of friction and wear on the surface and subsurface contact stresses generated—in an elastic half-space—by a cylindrical flat-ended punch, under cyclic indentation loading. For this purpose, the computational framework presented by the authors [71] was extended to study this fretting wear contact problem.

This numerical scheme allowed us to study simultaneously the evolution of the surface wear and the subsurface stress distributions (i.e., maximum value and its location) under orthotropic friction and wear conditions. This allowed us to evaluate the stress distributions in a wide range of engineering components, i.e., especially in those whose surfaces present some particular striation patterns (e.g., caused by machining operations), and highs and hollows were clearly oriented on the surface. For such cases, a specific orthotropic friction and wear law should be considered. However, the formulation presented some limitations. It should be applied to those domains where elastic half-space assumptions can be assumed; therefore, it is not valid for soft surfaces, where small strains cannot be assumed.

After the validation of this computational scheme, several numerical studies were developed to obtain the following conclusions:

- For the static normally loaded cylindrical flat-punch (see Section 7.1), the stick region in the contact zone increased with the value of the friction coefficient until a threshold value was reached (i.e., $\mu \approx 0.4$ in our analysis). Then, all the points of the contact area reached a contact–stick state. Additionally, on the subsurface von Mises stress distributions (see Figure 5b–f), we observed that the size of the region with $\sigma_{VM}/p_o \geq 1$ increased with the friction coefficient value—and new stress gradient around the $z$-axis appeared—until the threshold value was reached. However, when the friction intensity reached the mentioned threshold value, no more changes were observed in the normalized von Mises stress distribution.

- Regarding the response under isotropic fretting wear conditions caused by the normal cyclic loading conditions (i.e., radial fretting wear), we observed how the smaller the friction coefficient value was, the greater the wear damage (i.e., maximum wear depth or wear volume) obtained. This was due to the fact that the smaller the friction coefficient value was, the bigger the annular sliding contact zone obtained.

- The influence of friction intensity can be observed—after a big number of load cycles (i.e., $N \approx 10^5$)—on the resulting surface wear depth, the normal tractions, and the subsurface von Mises stress values. Due to the fact that isotropic friction and wear laws were considered, their distributions presented a $z$-axis symmetry. In particular, with regard to the location of the maximum stress value, the low friction values (i.e., $\mu = 0.1$) caused important wear damage on the solid surfaces. Moreover, this also led to a stress distribution where the maximum value of the subsurface von Mises stress was located at $r = 0$ and $z/a_o \approx 0.7$. On the other hand, when high friction values were considered (i.e., $\mu = 0.4$), low wear damage was produced on the solid surfaces, and therefore, the stress distribution presented its maximum value located on the surface $z/a_o \approx 0$ and at the edge of the contact circle ($r/a_o \approx 0.9$).

- Finally, the response under orthotropic fretting wear conditions caused by the normal cyclic loading conditions did not present a $z$-axis symmetry. The maximum value of the normal contact pressure distributions was located where the wear depth distributions presented the greatest gradient values. This region was located at the intersection of the edge of the contact circle and the direction of the tribological principal axis with the greatest friction coefficient. Moreover, contrary to what was observed for isotropic fretting wear conditions, the location of the maximum values of the subsurface stress after a high number of cycles ($N = 10^5$ cycles) remained at the edge of the contact circle, which, due to wear, was slightly less than $a_o$.

These conclusions revealed the following two main findings. First was the importance of the friction intensity in the evolution of the subsurface stress distributions (i.e., maximum value and its location) when a flat-ended cylindrical punch over an elastic half-space was

subjected to radial fretting wear conditions. Second was the significance of considering orthotropic friction and wear conditions in the evolution of the surface and the subsurface stress distributions for this problem.

**Author Contributions:** Conceptualization, J.M.J. and L.R.-T.; methodology, J.M.J. and L.R.-T.; software, J.M.J. and L.R.-T.; validation, J.M.J.; formal analysis, J.M.J.; investigation, J.M.J.; resources, J.M.J. and L.R.-T.; data curation, J.M.J.; writing—original draft preparation, J.M.J.; writing—review and editing, L.R.-T.; visualization, L.R.-T.; supervision, L.R.-T. All authors have read and agreed to the published version of the manuscript.

**Funding:** This research was funded by the *Consejería de Transformación Económica, Industria, Conocimiento y Universidades de la Junta de Andalucía* (Spain) through the research project P18-RT-3128.

**Data Availability Statement:** Not applicable.

**Conflicts of Interest:** The authors declare no conflict of interest.

## Nomenclature

Roman symbols:

| | |
|---|---|
| $a_o$ | Cylindrical punch radius |
| $A_{ij}(\mathbf{x}, \mathbf{x}')$ | Displacement kernel function to take into account the contribution of the surface $j$-component contact traction on the relative $i$-component displacement on the solid surface |
| $\mathbf{A}(\mathbf{x}, \mathbf{x}')$ | Displacement kernel function matrix |
| $\{e_1, e_2\}$ | Tribological axes |
| $E$ | Young's modulus |
| $E^*$ | Reduced or effective elastic modulus |
| $\lVert \bullet \rVert_\mu$ | Elliptic norm based on the friction intensity coefficient |
| $g_g$ | Geometrical gap |
| $\dot{g}_{e_1}$ | Tangential slip velocity ($e_1$-component) |
| $\dot{g}_{e_2}$ | Tangential slip velocity ($e_2$-component) |
| $g_n$ | Normal gap |
| $g_{n,o}$ | Rigid body normal approach |
| $g_{t,o}$ | Rigid body tangential slip |
| $\mathbf{g}_t$ | Tangential slip vector |
| $\dot{\mathbf{g}}_t$ | Tangential slip velocity vector |
| $G$ | Shear modulus |
| $i_w$ | Wear coefficient |
| $i_1$ | Principal wear coefficient in the $e_1$-direction |
| $i_2$ | Principal wear coefficient in the $e_2$-direction |
| $k_n$ | Contact stiffness |
| $N$ | Number of load cycles |
| $N_e$ | Number of mesh elements |
| $N_x$ | Number of mesh divisions in the $x$-direction |
| $N_y$ | Number of mesh divisions in the $y$-direction |
| $P$ | Static normal load |
| $p_{e_1}$ | Tangential contact traction ($e_1$-component) expressed in $\{e_1, e_2\}$ |
| $p_{e_2}$ | Tangential contact traction ($e_2$-component) expressed in $\{e_1, e_2\}$ |
| $p_o$ | Average contact pressure |
| $p_n$ | Normal contact pressure |
| $p_n^*$ | Augmented normal contact traction |
| $p_x$ | Tangential contact traction ($x$-component) |
| $p_y$ | Tangential contact traction ($y$-component) |
| $p_z$ | Normal contact traction |
| $\mathbf{p}$ | Contact traction vector |
| $\mathbf{p}_t$ | Tangential contact traction vector |
| $\mathbf{p}_t^*$ | Augmented tangential contact traction vector |
| $\mathbb{P}_{\mathbb{R}^+}(\bullet)$ | Normal projection function |

| $\mathbb{P}_{\mathbb{E}_\rho}(\bullet)$ | Tangential projection function |
|---|---|
| $r$ | Radial–cylindrical coordinate |
| $r_n$ | Normal penalty parameter |
| $r_t$ | Tangential penalty parameter |
| $T_{ij}^N(\mathbf{x}, \mathbf{x}')$ | Stress kernel function to take into account the contribution of the normal contact traction on the tress tensor $ij$-component in the solid |
| $T_{ij}^{S_x}(\mathbf{x}, \mathbf{x}')$ | Stress kernel function to take into account the contribution of the $x$-component tangential contact traction on the tress tensor $ij$-component in the solid |
| $T_{ij}^{S_y}(\mathbf{x}, \mathbf{x}')$ | Stress kernel function to take into account the contribution of the $y$-component tangential contact traction on the tress tensor $ij$-component in the solid |
| $\mathbf{T}(\mathbf{x}, \mathbf{x}')$ | Stress kernel function matrix |
| $u_n$ | Surface relative normal displacement |
| $u_x$ | Surface relative displacement ($x$-component) |
| $u_y$ | Surface relative displacement ($y$-component) |
| $u_z$ | Surface relative displacement ($z$-component) |
| $\mathbf{u}_t$ | Surface relative tangential displacement vector |
| $\{x, y, z\}$ | Cartesian coordinate system |
| $\mathbf{x}$ | Position vector of a solid interior point |
| $\mathbf{x}'$ | Position vector of a solid surface point |

Greek symbols:

| $\beta$ | Tribological axes' angle orientation |
|---|---|
| $\Gamma_c$ | Contact zone |
| $\Delta g_t$ | Tangential slip increment |
| $\Delta_x$ | Element size in the $x$-direction |
| $\Delta_y$ | Element size in the $y$-direction |
| $\Delta\tau$ | Pseudo-time increment |
| $\Delta\omega$ | Wear depth increment |
| $\mu$ | Friction intensity coefficient |
| $\mu_1$ | Principal friction coefficient in the $e_1$-direction |
| $\mu_2$ | Principal friction coefficient in the $e_1$-direction |
| $\nu$ | Poisson's ratio |
| $\sigma_{ij}$ | Stress tensor $ij$-component |
| $\sigma_{rr}$ | Radial stress |
| $\sigma_{\phi\phi}$ | Circumferential stress |
| $\sigma_{VM}$ | von Mises stress |
| $\sigma_{zz}$ | Normal stress |
| $\tau$ | Pseudo-time |
| $\tau_k$ | Pseudo-time $k$-instant |
| $\tau_{rz}$ | Shear stress |
| $\omega$ | Wear depth |
| $\dot{\omega}$ | Wear rate |
| $\Omega$ | Solid domain |

**Appendix A**

The stress components of an isotropic half-space in frictionless contact with a centrally loaded cylindrical flat punch can be expressed, in cylindrical coordinates—adapted from Kachanov et al. [98]—as a function of the radial distance ($r = \sqrt{x^2 + y^2}$) as

$$\sigma_{rr} = \frac{1}{2}(\sigma_1 + \sigma_2), \quad \sigma_{\phi\phi} = \frac{1}{2}(\sigma_1 - \sigma_2) \tag{A1}$$

$$\sigma_{zz} = \frac{P}{2\pi a_o}\left[-\frac{\sqrt{a_o^2 - l_1^2}}{l_2^2 - l_1^2} + \frac{z^2\left(l_1^4 + a_o^2(r^2 - 2a_o^2 - 2z^2)\right)}{\sqrt{a_o^2 - l_1^2}\left(l_2^2 - l_1^2\right)^3}\right], \tag{A2}$$

and

$$\tau_{rz} = -\frac{P}{2\pi a_o} \frac{zr\sqrt{a_o^2 - l_1^2}\left(3l_2^2 + l_1^2 - 4a_o^2\right)}{\left(l_2^2 - l_1^2\right)^3}. \tag{A3}$$

In the expressions above, $a_o$ is the punch radius, $P$ is the centrally applied vertical force,

$$\sigma_1 = -\frac{P}{2\pi a_o}\left((1+2\nu)\frac{\sqrt{a_o^2-l_1^2}}{l_2^2-l_1^2} + \frac{z^2\left(l_1^4 - a_o^2\left(2a_o^2 - r^2 + 2z^2\right)\right)}{\sqrt{a_o^2-l_1^2}\left(l_2^2-l_1^2\right)^3}\right), \tag{A4}$$

$$\sigma_2 = -\frac{P}{2\pi a_o}\left[(1-2\nu)\left(\frac{\sqrt{a_o^2-l_1^2}}{l_2^2-l_1^2} - \frac{2\left(a_o - \sqrt{a_o^2-l_1^2}\right)}{r^2}\right) + \frac{a_o z\sqrt{l_2^2-a_o^2}\left(2l_1^4 + r^2\left(l_1^2 + 3l_2^2 - 6a_o^2\right)\right)}{l_2^2\left(l_2^2-l_1^2\right)^3}\right], \tag{A5}$$

$$l_1 = \frac{1}{2}\left(\sqrt{(r+a_o)^2 + z^2} - \sqrt{(r-a_o)^2 + z^2}\right), \tag{A6}$$

and

$$l_2 = \frac{1}{2}\left(\sqrt{(r+a_o)^2 + z^2} + \sqrt{(r-a_o)^2 + z^2}\right). \tag{A7}$$

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
