# Peer review of "Wear and Subsurface Stress Evolution in a Half-Space under Cyclic Flat-Punch Indentation"

_lubricants, doi:10.3390/lubricants11060265_

Round 1

Reviewer 2 Report

This manuscript analyzes the influence of friction and wear on the surface and sub-surface contact stresses generated –in an elastic half-space– by a cylindrical flat-ended punch, under cyclic indentation loading. They also carried out a comparison between the theoretical normal contact pressure distribution and the numerical results, they match so well. Overall, the article is well organized, and its presentation is good. Hence, I recommend this article for publication in the journal lubricants.

Reviewer 3 Report

1- Certain important observations have not been made in the paper. Therefore, the Authors should note that failure by induced sub-surface stresses occurs when these stresses exceed elastic limits according to given/pertinent failure criteria.

The failure is usually due to fatigue spalling/pitting . It is also important to note that failure occurs when such limiting stresses coincide with sub-surface flaws created in manufacture such as pores, inclusions, cracks, etc.

2-        Citations should not be included in the Abstract, i.e. “[1]”.

3-      A claim to originality should be briefly included in the Abstract, i.e, what is new in this paper? (please see points 5 and 6 below, before a claim of originality is made).

4-      The claim to original contribution should be expanded upon by a final paragraph in the paper’s Introduction.

5-      Line 47  of Introduction states that “Previous works are mainly focused on the surface analysis (i.e., the evolution of the 47 surface wear and contact tractions) in non-conformal contact problems under sliding wear 48 or fretting wear conditions.” This sentence is factually incorrect. Please see the following paper which should be cited and the claim made toned down:

Johns-Rahnejat, P.M., Dolatabadi, N. and Rahnejat, H., “Analytical elastostatic contact mechanics of highly-loaded contacts of varying conformity”, Lubricants, 2020, 8(9):89. 

6-      The review of literature in determining sub-surface stress field and potential failure by fatigue is rather poor and misses a fair number of important references. These include the following, which should be included, before any claim to originality is made in points 3 and 4 above:

Johnson, K.L., “Contact mechanics and the wear of metals”, Wear, 1995, 190(2), pp. 162-170.

Johns-Rahnejat, P.M. and Gohar, R., “Point contact elastohydrodynamic pressure distribution and sub-surface stress field”,  In Tri-annual conference on multi-body dynamics: monitoring and simulation techniques,  Mechanical Engineering publishing, Bradford, 1997.

Teodorescu, M., Rahnejat, H., Gohar, R. and Dowson, D., “Harmonic decomposition analysis of contact mechanics of bonded layered elastic solids”, Applied Mathematical Modelling, 2009, 33(1), pp. 467-485.  

Greenwood, J.A., “Subsurface stresses in an elliptical Hertzian contact”, The Journal of Strain Analysis for Engineering Design, 2022, 57(8), pp. 677-687

May be the BEM approach makes and combined wear and fatigue should be claimed as the original approach.

7-     The authors should make is clear that von Mises failure Criterion only applies to ductile contacting bodies, where semi-infinite elastic assumption can also be assumed.

The same is not true for hardened surfaces or hard wear-resistant coatings such as ceramics. For layered surfaces the readers should be referred to the work of Teodorescu et al above.

8-     The methodology developed in this paper is only valid for harder surfaces, not sift surfaces where small strains cannot be assumed. This should also be stated.

9-     Practical implications of the results should be stated in the paper’s Conclusions.

10-  Please also state what is the major fond of the paper in its conclusion.

11-  There should be a full nomenclature of all the mathematical symbols used in alphabetic order.

I look forward to receiving any revised version, with all the changes made clearly highlighted. Please also respond to the issues raised above in a point-by-point basis.

The English grammar is generally fine.

Round 2

Reviewer 1 Report

All my comments are satisfactorily responded. 

Reviewer 3 Report

The authors have significantly improved their paper. I recommend the revised version for publication.

The quality of English is overall fine, but some copy-editing at the Proofs-stage would benefit the paper.